

# Biodiversity and temporal patterns of macrozoobenthos in a coal mining subsidence area in North China

Guanxiong Zhang[1,2,3], Xingzhong Yuan[1,2,3] and Kehong Wang[1,2,3]

[1] Chongqing University, State Key Laboratory of Coal Mine Disaster Dynamics and Control, Chongqing, China
[2] Chongqing Key Laboratory of Wetland Science Research in the Upper Reaches of the Yangtze River, Chongqing, China
[3] Institute of Chongqing Qianzhou Ecological Research, Chongqing, China

## ABSTRACT

Coal resources play a strategic role in the long-term development of China. Large-scale mining has a considerable impact on the landscape, and it is a long-term heritage of industrialization unique to the Anthropocene. We investigated the macrozoobenthos and water in nine mining subsidence wetlands at different developmental stages (3–20 years) in North China. A total of 68 species were found, and the macrozoobenthos community in the newly formed wetlands showed high diversity. We believe that this high diversity is not random; rather, the high diversity was because of the special origin and development of the wetland. We used three time slices from the timeline of the development of the newly formed wetlands and compared them. It was found that the macrozoobenthos community was significantly affected by the change in the subsidence history. We emphasize that coal mining subsidence should not be merely identified as secondary man-made disasters, as they are often secondary habitats with high conservation value, and their conservation potential lies in the fact that these secondary habitats can replace rapidly decreasing natural wetlands.

## INTRODUCTION

Coal resources play a strategic role in the long-term development of China. The rapid development of the social economy has resulted in an increased demand for energy. Thus, coal mining activities have become more intense. In 2014, coal production reached 38.7 billion tons, accounting for 49% of the world's gross domestic product (The National Bureau of Statistics of the People's Republic of China); of this, more than 92% of the coal comes from underground mining, which leads to serious surface subsidence. It has been estimated that when 10,000 tons of coal is mined underground, the settlement area spans 0.20–0.33 hectares. In China, the annual subsidence is expected to increase by 2,104 ha (*Hu & Xiao, 2013*). High groundwater coal basins are mainly distributed in the eastern and northeast parts of China. There are many thick coal seams with high groundwater levels and flat terrain. Most of these areas overlap coal seams and agricultural

Corresponding author
Xingzhong Yuan,
1072000659@qq.com

production sites. After coal mining, the upper roof of the mine is broken, resulting in changes in the original geological structure; this process further induces ground subsidence, and a large amount of mine subsidence water covers areas and has high depths, that is, up to 13 m in some places. The groundwater will appear if the basin surface subsides to the phreatic water surface (*Zeng et al., 2016*). After coal mining subsides, the subsidence pond forms a subsidence lake. Under flooding conditions, submerged soil gleying occurs, wetland plants are formed, and typical "mining subsidence wetlands" (MSWs) are formed.

In stark contrast to valuable habitats such as natural wetlands, coal mining subsidence often reforms the environment, greatly limiting environmental recovery. In this case, it is impossible to maintain the original environmental attributes or to maintain the native species pools, which means that the succession process of the organism basically starts from an initial (or primitive) stage (*Suding, Gross & Houseman, 2004*). However, some studies have reported that the temporal structural changes of macrozoobenthos (*Layton & Voshell, 1991*; *Christman & Voshell, 1993*; *Coccia et al., 2016*) in newly formed ponds reach a high density several months after the ponds are filled, and macrozoobenthos become an important food source for benthic fish (*Solimini et al., 2003*). However, research on the macrozoobenthos community in coal mining subsidence areas is scarce.

Macrozoobenthos, as a potentially important driver of metabolism in freshwater ecosystems, have traditionally been considered a food source for benthic fish and consumers of algae and organic matter (*Janse et al., 2010*; *Vander Zanden & Gratton, 2011*). Specifically, macrozoobenthos provide an important link between primary producers and secondary consumers, play an important role in the nutritional cycle, and promote the decomposition of organic matter through consuming and decomposing plant and animal tissues (*Irons, Oswood & Bryant, 1988*). This community is also an ideal biological indicator of the aquatic ecosystem (*Mereta et al., 2013*; *Chang et al., 2014*; *Hong et al., 2014*) because of the following characteristics (*Burgmer, Hillebrand & Pfenninger, 2007*; *Schilling, Loftin & Huryn, 2009*; *Villnäs & Norkko, 2011*; *Kappes & Haase, 2012*; *Hölker et al., 2015*): diverse species, poor swimming ability, sensitivity to environmental change and ease of collection.

It is generally assumed that the degree of anthropogenic influence is negatively correlated with the health of the macrozoobenthos community. In industrial and post-mining areas, the lower biodiversity is assumed to be related to the surrounding natural habitats (*Baasch, Kirmer & Tischew, 2012*; *Hendrychová et al., 2012*; *Pérez-Bilbao, Benetti & Garrido, 2015*; *Podgorska, 2015*). Man-made (i.e., secondary) habitats can increase the diversity of habitats within the local area and can replace the original habitat. There are a large number of studies showing that human disturbance on natural habitats has a negative impact on the taxonomic diversity, but it is surprising that environmental changes caused by human activities may indirectly lead to the establishment of heterogeneous habitats (*Samways, 1989*; *Tropek et al., 2010*; *Dolny & Harabis, 2012*; *Chester & Robson, 2013*). A large number of heterogeneous patches (habitats) have developed because of the direct or indirect results of surface (*Tropek et al., 2010*) and sub-surface mining (*Dolny & Harabis, 2012*). These habitats are often

regarded as local diversity hotspots for terrestrial and freshwater invertebrates, including threatened and endangered species (*Benes, Kepka & Konvicka, 2003*; *Tropek et al., 2010*; *Dolny & Harabis, 2012*).

For some species, post-mining habitats are the only suitable habitat within a vast area (*Harabis & Dolny, 2015*). It has been speculated that individual species have different sensitivities to the characteristics of individual habitats and that they can be discovered at a particular location only when the environment meets their unique habitat requirements (*Lubertazzi & Ginsberg, 2009*; *Ball-Damerow, M'Gonigle & Resh, 2014*). Increasingly, research indicates that the initial environmental characteristics may affect the biodiversity in a region and that the initial characteristics of the newly formed environment may have a significant impact on the subsequent succession processes (*Prach & Pyšek, 2001*). Therefore, from the perspective of environmental conservation, we are eager to understand these processes and initial habitat characteristics, which may have a significant impact on the biodiversity and conservation value of newly developed wetlands in coal mining subsidence areas. The purpose of this study was to determine the general pattern of macrozoobenthic communities in freshwater wetlands in coal mining subsidence areas. We designed four research purposes in combination with data: (1) studying the species composition and diversity of the macrozoobenthic community in MSWs; (2) studying the temporal pattern of macrozoobenthos at different developmental stages in MSWs; (3) assessing the impact of environmental variables on the diversity and conservation of these secondary habitats; and (4) proposing appropriate management recommendations based on the synthesis of this knowledge.

## MATERIALS AND METHODS

### Study area

The study area is located in Yanzhou City (116°50′–116°55′E, 35°30′–35°25′N), Shandong Province, which is an important coal production base in China (Fig. 1). Shandong Province's climate is typical temperate monsoon climate characterized by hot and rainy summers and cold and dry winters, with an average annual precipitation of 600 mm and a mean temperature between 13.5 and 15 °C (*Yu et al., 2017*). Rapid economic development and a considerable increase in population have triggered the evident expansion of urban areas in China, including Yanzhou City was no exception. The direct results of this expansion have included a decrease in natural wetlands from 1985 to 2015 (*Xiao et al., 2018*). Historically, the region is characterized by agriculture, and the modern agricultural economy is concentrated on the cultivation of corn and wheat. Underground coal mining has been performed in this region for more than 40 years. The ground has been moved, deformed and even destroyed, forming a large area of underground mined-out areas, which eventually formed a large-scale subsidence. Due to the shallow groundwater and a large amount of rainwater inflow, soil paludification has formed small closed-type new wetlands of varying sizes in this region. These wetlands have a short formation time, are relatively isolated from one another, are surrounded by farmland, have no other hydrological connection, have high water alkalinity, have complex chemical composition, and have a sediment type classified as sludge.

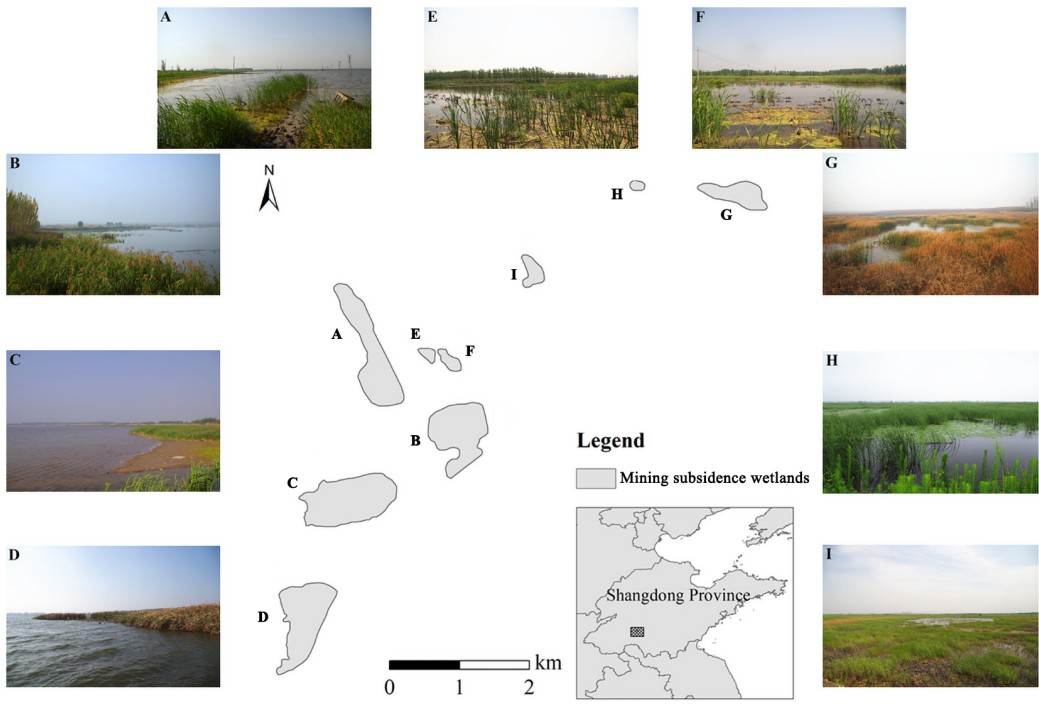

**Figure 1** **The map shows study area is located in Shandong Province, China, and (A, E, F) shows the initial stage of mining subsidence wetlands (MSWs), and (G, H, I) show the middle stage of MSWs; the remaining photographs depict (B, C, D) the late stage of MSWs.** Photo credits: Guanxiong Zhang.

These wetlands are similar to natural shallow lakes, with water surface areas ranging from 0.59 to 60.51 ha. The main hydrological effects come from rainfall and groundwater. The wetland plants are dominated by emergent plants, such as *Phragmites australis* and *Typha angustifolia*. The main land-use pattern around the wetland is farmland. The crops are dominated by corn in summer and autumn and by wheat the rest of the year.

## Sampling and sample processing

Newly formed wetlands without anthropogenic interference were the main focus in the current study, and nine wetlands with subsidence periods between 2 and 20 years were selected as targets (Fig. 1). We divided the wetlands into three groups according to the history of subsidence, named the initial stage (IS, three wetlands with a subsidence history of 2–5 years), the middle stage (MS, three wetlands with a subsidence history of 9–11 years), and the late stage (LS, three wetlands with a subsidence history of 18–20 years).

The three subsidence time phases correspond to the three developmental stages of the new wetland:

(1) IS. In the early stage of the formation of fresh wetlands, the wetland area is small, the environment is fragile, and the water depth is shallow (<1 m). The wetland water volume is greatly affected by precipitation. In years with long-term drought, dryness will occur. The aquatic plants are mainly *Phragmites australis* and *T. angustifolia*, and no fish distributions were found in the wetlands during the study period.

(2) MS. Due to the continuous subsidence of the Earth's surface, the newly developed wetland is in a dynamic and repetitive developmental stage. After nearly 10 years of development, the area becomes enlarged, and the water depth increases (one to two m). The main aquatic plants are *Phragmites australis* and *T. angustifolia*, and a small amount of fish were found during the research period (e.g., *Monopterus albus, Misgurnus anguillicaudatus*).

(3) LS. In MSWs with a subsidence time of approximately 20 years, the subsidence process ends and the wetland area is further expanded to form a subsidence lake with a maximum depth of five m. Additionally, broad coastal vegetation, composed of *Phragmites australis* community and *T. angustifolia* community, is present. Both of these are single-species communities, reducing the diversity of vegetation composition. The surface sediment contains a large amount of organic detritus, and four fish species were found in these wetlands during the study (e.g., *Monopterus albus, Misgurnus anguillicaudatus, Carassius auratus*, and *Ctenopharyngodon idellus*).

The samplings were conducted in October 2016 (autumn), January 2017 (winter), April 2017 (spring) and July 2017 (summer). Environmental variables were collected in parallel in April and July 2017 (Table 1).

## Macroinvertebrate sampling

We established three sites in each MSW, with the average water depth ranging from 0.4 to 1.5 m. The bottoms of the sites were mainly characterized by sediments. Samples were collected three times from each site (collected from shallow to deep) using a 1/40 Peterson grab sampler and then combined as one sample. The total amount of sampling effort was equal at all sites. The collected samples were sorted using a 250-μm pore size mesh, placed in a sealed bag, and stored in 7% buffered formalin. The collected samples were classified under the microscope and identified to the lowest possible classification unit. Mollusca and Oligochaeta were identified to species level, and taxa that could not be identified to species level (e.g., most species of Diptera, Trichoptera, Odonata, and Ephemeroptera) were identified to morphospecies. The identification data used were mainly those described by *Rosenberg & Resh (1994)*. We summed the number of each classification unit and then used filter paper to remove the surface solution. The units were weighed on an electronic balance, and the final result was converted into density and biomass per unit area. Macrozoobenthos were collected under the field permit approval (Foundation of State Key Laboratory of Coal Mine Disaster Dynamics and Control, Chongqing University) and the project number: 2011DA105287—ZD201402.

## Water characteristics

The physicochemical parameters of the surface water, including dissolved oxygen (DO), water temperature (WT), total dissolved solids (TDS), salinity (Sal), oxidation reduction potential (ORP), and pH, were measured using the calibrated Manta™ 2 Multiparameter System (Eureka Company, Medford, MA, USA). A vertically integrated water sample was also collected and placed in an acid-cleaned plastic container with a

**Table 1 Environmental variables describing the study sites.**

| Environmental variables | Min–Max (Mean) | | | Units |
|---|---|---|---|---|
| | Initial stage | Middle stage | Late stage | |
| Subsidence history | 3–5 (5) | 9–11 (10) | 18–20 (19) | Year |
| Water area | 3.09–30.17 (5.82) | 1.84–11.85 (6.88) | 32.86–60.51 (50.63) | ha. |
| Water depth | 0.3–0.7 (0.5) | 0.3–1 (0.6) | 0.4–1.8 (0.8) | m |
| Water temperature | 18.66–30.78 (27.08) | 20.49–29.38 (26.42) | 20.9–31.73 (27.94) | °C |
| Chlorophyll-*a* | 1.09–30.17 (5.82) | 0.55–41.50 (6.55) | 0.55–134.86 (18.29) | μg/L |
| Total dissolved solids | 6.29–15.78 (12.26) | 5.10–12.45 (9.20) | 11.10–14.51 (12.60) | g/L |
| Salinity | 5.42–14.78 (11.18) | 4.33–11.91 (8.18) | 9.96–13.41 (11.50) | % |
| Dissolved oxygen | 0.75–12.48 (5.59) | 0.13–14.03 (3.74) | 3.25–11.94 (7.01) | mg/L |
| pH | 6.17–8.66 (7.75) | 5.50–7.46 (6.99) | 6.03–9.53 (8.16) | – |
| Oxidation reduction potential | 39.6–288.3 (127.0) | 13.6–265.3 (148.2) | 90.7–300.7 (182.6) | mV |
| Total phosphorus | 0.42–0.63 (0.18) | 0.09–0.52 (0.13) | 0.08–0.52 (0.15) | mg/L |
| Total nitrogen | 0.82–2.13 (1.52) | 0.70–2.41 (1.14) | 0.90–2.41 (1.45) | mg/L |

volume of five L. The total nitrogen (TN), total phosphorus (TP), and chlorophyll-*a* (*Chl-a*) were measured in the laboratory according to APHA (2012).

## Functional feeding groups

According to the food source and feeding mechanism (*Barbour et al., 1999*), macrozoobenthos were categorized into five functional feeding groups (FFGs): filter collector (FC), predator (PR), shredder (SH), gatherer collector (GC), and scraper (SC). In the current study, the FC group accounted for only a small proportion; therefore, the FC group was eliminated from the study.

## Biodiversity metrics

We calculated the macroinvertebrate Shannon–Wiener diversity index ($H'$) using the Excel and Biodiversity tools. SPSS was used for statistical analysis.

Shannon–Wiener diversity index ($H'$):

$$H' = -\sum_{i=1}^{S} Pi \ln Pi$$

where $Pi$ is the proportion of the total sample represented by genus ($S$) $i$.

## Data analyses

First, permutational multivariate analysis of variance was used to determine the effects of subsidence time on the community composition of macrozoobenthos (999 permutations). The relationships between the macrozoobenthos community and the environmental factors (i.e., pH, TN, TP, *Chl-a*, area, WT, TDS, DO, ORP, and Sal) were explained by canonical correspondence analysis (CCA) (*Ter Braak & Smilauer, 2002*). Each environmental factor (except pH) and the relative abundance (≥1‰) were converted ($\log(x+1)$) before analysis. Both analyses were based on the Bray–Curtis similarity index of

the relative abundance and environmental data (pH excluded) after the log($x$+1) transformation. The rare taxa (<1%) were removed before analysis. The significance of the influence of each environmental factor on the community composition was analyzed with a Monte Carlo test, and the contribution of the Bonferroni correction was based on the conditional effect of the CCA result.

Second, general linear models (GLMs) were used to test the subsidence time on the total richness, density, biomass, and $H'$ of the macrozoobenthos communities, as well as on the richness and the relative abundance of the four FFGs and the major dominant taxa. Data were log($x$+1) transformed when they were abnormally distributed. When significant differences were identified from the GLMs, post hoc analysis was conducted for multiple comparisons by Tukey's honestly significant difference test. Before the analyses, the richness, density, biomass, and $H'$ were averaged from three ponds and then averaged from the four sampling seasons. Additionally, the seasonal dynamics of the richness, density, biomass, and $H'$ of the macrozoobenthos communities from the three stages were also analyzed.

## RESULTS

### Macrozoobenthos community composition

A total of 108 samples, including 68 taxa of macrozoobenthos and 15,332 individuals, were collected in the current study (see Appendix). The main group was aquatic insects, including 48 species, followed by molluscs (14 species). Furthermore, Chironomidae dominated all the sites and was the most taxonomically rich family (24 species).

Permutational multivariate analysis of variance showed that the community composition of the macrozoobenthos in the MSWs varied significantly with the time of subsidence ($F$ = 8.1114, $P$ < 0.001), significant differences were found between IS and LS and between MS and LS ($F$ = 8.336, $P$ < 0.01; $F$ = 9.541, $P$ < 0.01, respectively), while there was no significant difference between IS and MS ($F$ = 1.602, $P$ > 0.05).

In each stage of the MSWs, the main group included aquatic insects (Fig. 2). Additionally, the relative abundance of aquatic insects showed a tendency to decrease as the subsidence time increased, whereas Oligochaeta exhibited the opposite tendency, that is, its relative abundance showed a tendency to increase as the subsidence time increased. In MS, Mollusca showed a higher percentage than other stages of the MSWs.

In the IS and MS of the MSWs (Table 2), the dominant species were *Chironomus plumosus*, *Glyptotendipes* sp. and *Einfeldia* sp.; LS was different from IS and MS, at this stage, the dominant species were *Limnodrilus hoffmeisteri*, *Chironomus plumosus*, and *Propsilocerus akamusi*.

### Macrozoobenthos community structure

ANOVA showed that (Fig. 3) the richness, biomass and $H'$ of macrozoobenthos in MSWs varied significantly with the time of subsidence ($F$ = 18.478, $P$ < 0.001; $F$ = 16.503, $P$ < 0.001; and $F$ = 7.592, $P$ < 0.01, respectively); all three of them were highest in MS. Significant changes in richness and $H'$ were recorded in LS ($P$ < 0.05). Significant changes

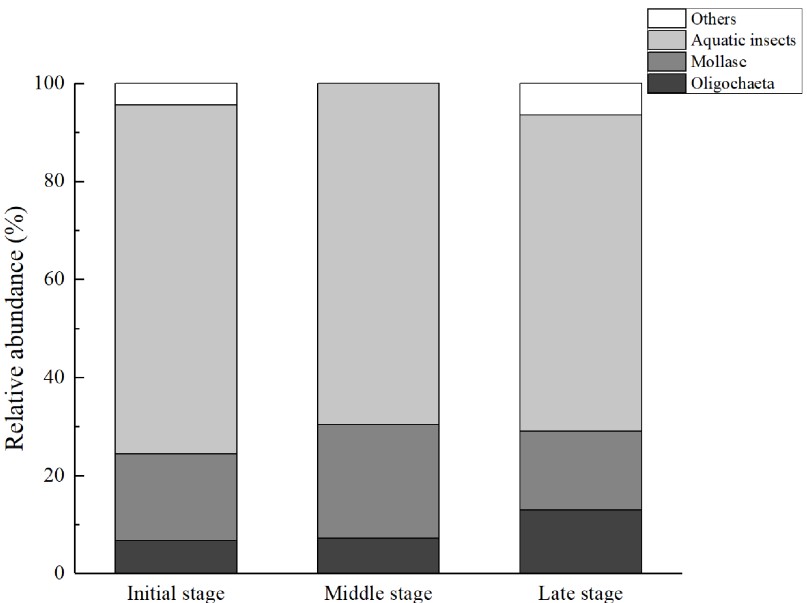

**Figure 2 Mean the community composition of the macrozoobenthos of each stage in mining subsidence wetlands.**

**Table 2 The dominant species of the three stages of the mining subsidence wetlands.**

| Subsidence history | Dominant species (relative abundance) |
|---|---|
| Initial stage | *Chironomus plumosus* (35.80%), *Glyptotendipes* sp. (34.50%), *Einfeldia* sp. (11.32%) |
| Middle stage | *Glyptotendipes* sp. (36.83%), *Chironomus plumosus* (17.86%), *Einfeldia* sp. (6.94%) |
| Late stage | *Limnodrilus hoffmeisteri* (39.35%), *Chironomus plumosus* (24.18%), *Propsilocerus akamusi* (14.11%) |

in biomass were recorded only in MS ($P < 0.05$). The density was not significantly affected by the subsidence time ($F = 1.862$, $P > 0.05$), which showed a tendency to decrease as the subsidence time increased ($P > 0.05$).

## Functioning feeding groups

ANOVA showed (Fig. 4) that the richness of SH, SC, and PR ($F = 7.636$, $P < 0.01$; $F = 17.856$, $P < 0.001$; $F = 6.812$, $P < 0.001$, respectively) in the MSWs varied significantly with the time of subsidence, and GC ($F = 0.832$, $P > 0.05$) in the MSWs was not significantly affected by the subsidence time. The highest richness of the four FFGs was recorded in the MS, and the lowest in the LS.

The relative densities of GC, SH, and SC ($F = 5.623$, $P < 0.01$; $F = 3.325$, $P < 0.05$; $F = 6.051$, $P < 0.01$, respectively) in the MSWs varied significantly with the time of subsidence, and PR ($F = 0.005$, $P > 0.05$) in the MSWs was not significantly affected by the subsidence time. Significant changes in both GC and SH were recorded in LS ($P < 0.05$). Significant changes in SC occurred only between MS and LS, where relative densities in MS were significantly higher than those in LS.

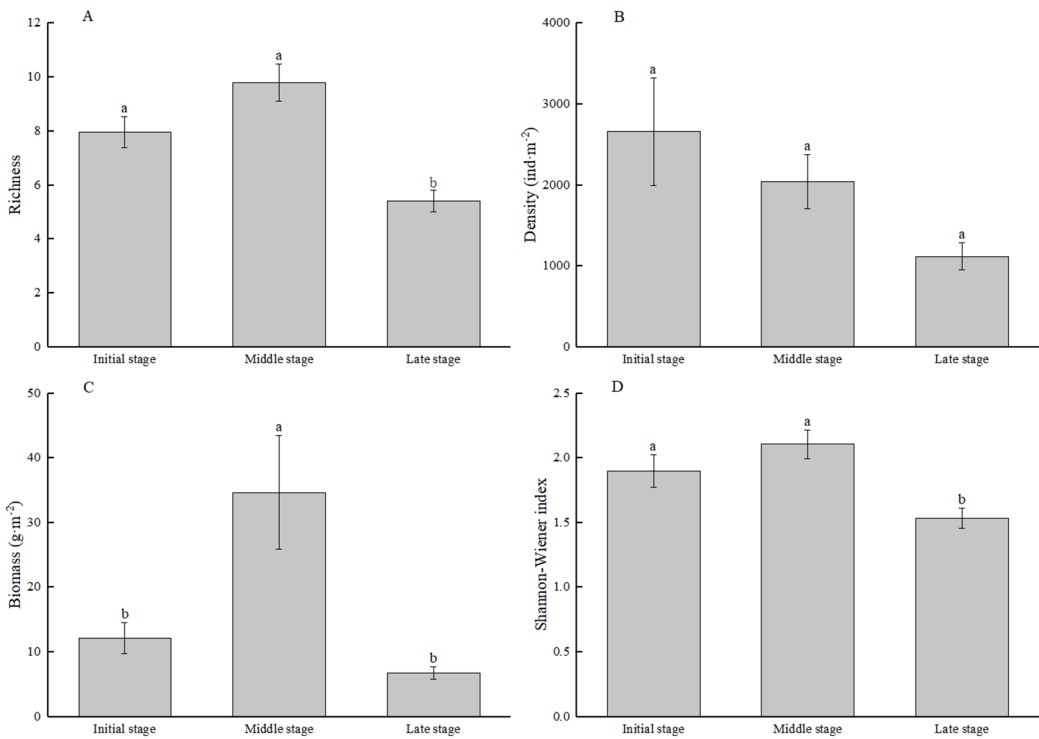

**Figure 3 Mean species richness (A), density (B), biomass (C), and Shannon–Wiener index (D) of macrozoobenthos of each stage in mining subsidence wetlands.** Standard error is represented by the vertical bars. Different letters represent significant differences in macrozoobenthos communites based on Tukey HSD test.

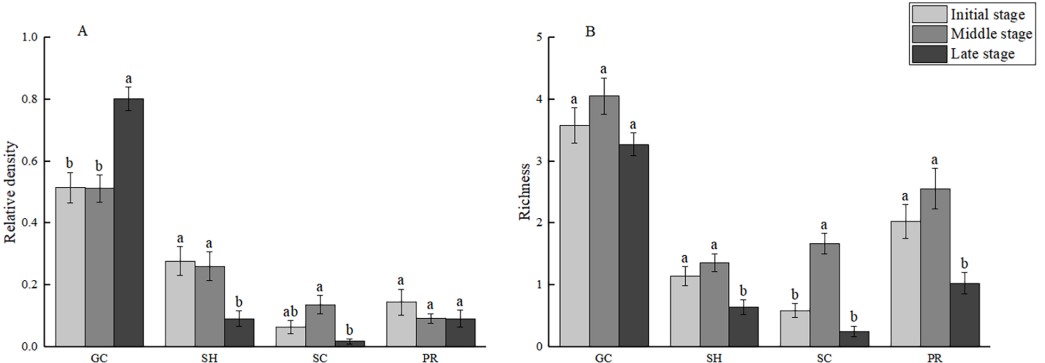

**Figure 4 Mean species relative density (A) and richness (B) of functional feeding groups (FFGs) of macrozoobenthos of each stage in mining subsidence wetlands.** Standard error is represented by the vertical bars. Different letters represent significant differences in macrozoobenthos communites based on Tukey HSD test. Abbreviations: GC, gatherer collector; SH, shredder; SC, scraper; PR, predator.

## Seasonal dynamics

The results of the GLMs (Fig. 5) showed significant seasonal differences in richness, density, biomass ($F = 4.555$, $P < 0.001$; $F = 7.178$, $P < 0.001$; and $F = 2.622$, $P < 0.05$, respectively), whereas there was no significant difference in $H'$ ($F = 0.882$, $P > 0.05$). However, the richness and density in MS and IS showed apparent

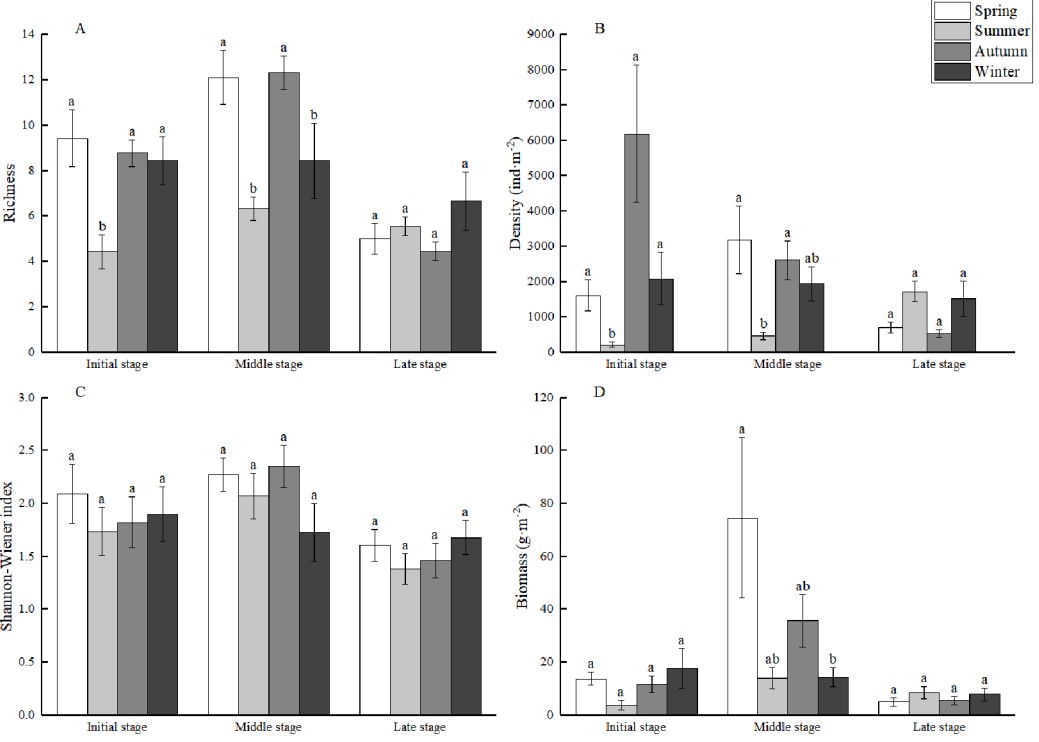

**Figure 5 Mean species richness (A), density (B), Shannon–wiener index (C), and biomass (D) of each stage in mining subsidence wetlands during the four seasons of collection.** Standard error is represented by the vertical bars. Different letters represent significant differences in macrozoobenthos communites during the four seasons of collection based on Tukey HSD test.

seasonal variations ($P < 0.05$), which was different from the results of LS. The lowest richness and densities in IS and MS were recorded in summer. Significant changes in biomass were recorded only in MS, and the highest biomass was recorded in spring.

## Relationship between macrozoobenthos community and environmental factors

The relationship between the macrozoobenthos and environmental factors in the MSWs was analyzed using CCA ($F = 2.000$, $P < 0.01$) (Fig. 6). The eigenvalues of the first two sorting axes in the sorted graph were 0.4227 and 0.3185, respectively. The correlation coefficients of the first two axes with the environmental factors were 0.8543 and 0.7787, respectively, and the environmental factors explained a total of 23.10% of the species variation information. Therefore, it can be seen that ORP, area, DO, and TDS were the main environmental factors affecting macrozoobenthos in the MSWs, explaining 22.51%, 19.48%, 16.88%, 16.88% of the total variance, respectively.

## DISCUSSION

Some specific secondary habitats (such as MSWs) can provide enormous biodiversity conservation potential (*Chester & Robson, 2013*; *Harabiš, Tichanek & Tropek, 2013*; *Harabiš, 2016*). In this particular type of coal mining subsidence, we found that

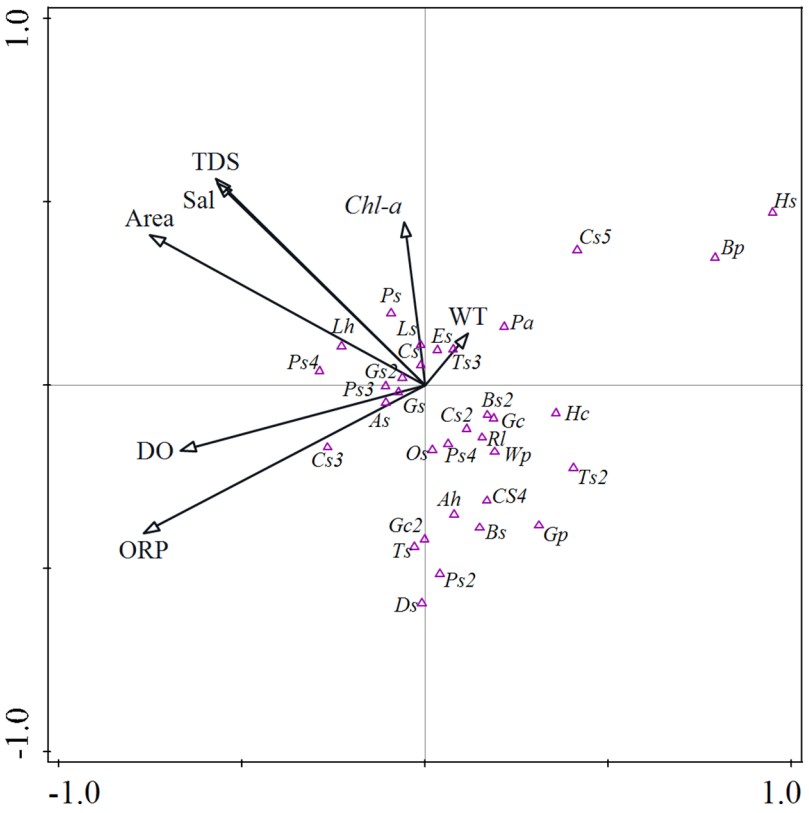

**Figure 6 Canonical correspondence analysis (CCA) ordination depicting relationships among abundances of macrozoobenthos and environmental variables.** Macrozoobenthos abbreviations: Ah, *Aciagrion hisopa*; Bp, *Bellamya purificata*; Bs, *Branchiura sowerbyi*; Bs2, *Baetis* sp.; Cs, *Chironomu splumosus*; Cs3, *Cricontopus* sp; Cs2, *Cladopelma* sp.; Cs4, *Cercion* sp.; Cs5, Corixidae sp.; Ds, *Dicrotendipes* sp.; Es, *Einfeldia* sp.; Hc, *Hippeutis cantori*; Hs, *Hippeutis cantori*; Gc, *Gyraulus convexiusculus*; Gc2, *Gyraulus compressus*; Gs, *Glyptotendipes* sp.; Gs2, *Gomphus* sp.; Gp, *Galba pervia*; Lh, *Limnodrilus hoffmeisteri*; Ls, *Libellula* sp.; Os, *Orthocladius* sp.; Pa, *Physa acuta*; Ps, *Propsilocerusi* sp.; Ps2, *Parafossarulus striatulus*; Ps3, *Polypedilum* sp.; Ps4, *Procladius* sp; Ps5, *Phryganea* sp1.; Rl, *Radix lagotis*; Ss, *Somatochlora* sp.; Ts, *Tvetenia* sp.; Ts2, *Tanytarsus* sp.; Wp, *Whitmania pigra*.

the developmental trajectory of the macrozoobenthos community in the MSWs was significantly affected by the time-lapse of the subsidence, which was driven by the advancement of the subsidence process. At the same time, our study also found that there was no significant difference in the macrozoobenthos community during the continuous succession period of IS and MS, indicating that the impact of subsidence time on the macrozoobenthos community was limited in a short period of time.

## Diversity in MSWs

In post-mining areas, there could be high heterogeneity in secondary habitats compared with the surrounding landscapes (*Tropek et al., 2010*). Therefore, macrozoobenthos generally exhibit higher diversity in post-mining landscapes. For example, 35 gastropod species were found in coal mining subsidence sites in the Silesia Highlands of southern Poland (*Strzelec, Krodkiewska & Krolczyk, 2014*); similarly, 40 odonate species were found in coal mining subsidence areas in the Czech Republic (*Harabiš, 2016*).

Therefore, we believe that the high diversity of this habitat is not random; rather, the high diversity depends on the particular origin and environmental heterogeneity caused by subsequent succession processes, which are a direct result of mining.

The macrozoobenthos of the MSWs are composed of ubiquitous species, and most species prefer natural habitats at low altitudes. However, massive natural habitats are disappearing due to urban expansion (*Mao et al., 2018*; *Xiao et al., 2018*). Therefore, we believe that secondary habitats, such as MSWs, are another habitat for lowland (plain) species, and these habitats are similar to other secondary alternatives to natural habitats, especially in landscapes that are strongly affected by human activities (*Céréghino et al., 2008*; *Le Viol et al., 2009*; *Chester & Robson, 2013*). Our research focused on the typical agricultural landscape of the North China Plain, which has relatively few natural habitats. There is only one type of similar hydrostatic water, which is often used in large volumes for fish farming. These aquaculture operations are homogeneous habitats, that is, they have lower value for biodiversity conservation than that of MSWs. This means that the only alternatives to ponds are secondary habitats that result from coal mining activities. Therefore, we must ask: what are the factors that provide higher diversity in MSWs?

## Macrozoobenthos community composition and diversity changes

Although we were unable to immediately measure the macrozoobenthos after the formation of new wetlands in coal mining subsidence areas, the rapid colonization of macrozoobenthos in the MSWs was predictable because this phenomenon is very common in newly created ponds (*Ruhí et al., 2009*; *Thiere et al., 2009*; *Ishiyama, Akasaka & Nakamura, 2014*). In fact, the newly emerged water pools in post-mining landscapes often provide particularly important refuges for species specialized for oligotrophic conditions and early-successional freshwater habitats (*Dolny & Harabis, 2012*; *Harabiš, Tichanek & Tropek, 2013*; *Harabiš, 2016*). Most of the increased macrozoobenthos richness was due to the immigration of active dispersers such as chironomids, beetles and dragonflies. In our study, aquatic insects accounted for more than 50% of all stages of development in newly developed wetlands, of which Chironomidae was the main component (Fig. 2). This result suggests that pioneer taxa indiscriminately colonized the newly developed wetlands at different stages of development, irrespective of their environmental differences. Due to their special life history, chironomids can move and reproduce in different areas and can complete colonization early in the formation of wetlands. Chironomidae adults are effective dispersers (*Batzer & Wissinger, 1996*), and they are usually the first colonizers to reach new wetlands (*Barnes, 1983*; *Layton & Voshell, 1991*). Due to the lack of natural enemies such as fish in newly formed wetlands, the distribution of chironomid larvae showed a higher density than those of other groups. Chironomidae rapidly colonized the newly formed wetlands in the coal mining subsidence area. Most of them belonged to the GC in the FFG classification, indicating that the newly formed wetlands had sedimentary environmental conditions with fine sediment deposition and high organic content. This result is consistent with the land use

(i.e., farmland) before the formation of the newly formed wetlands (*Xiao et al., 2018*). However, some taxa, such as Gastropoda, were absent, and the presence of some taxa, such as Oligochaeta, showed low abundance, suggesting they had not yet established during the IS.

In our study, the diversity of macrozoobenthos increased during the succession period from IS to MS, while the diversity of LS was significantly lower than that of the first two stages. Therefore, another question surfaced from these results: Why does the biodiversity of macrozoobenthos in MSWs decrease over time? One possible explanation is that the elderly wetlands had not yet been colonized by some taxa. The early colonists are usually too generalist to be affected by certain habitat attributes (*Jocque, Vanschoenwinkel & Brendonck, 2010*), and some specific species are found in longer time segments. The subsidence history of MS is longer than that of IS, and the development of habitat may allow the colonization of new species. The increase in the richness of molluscs in MS may be due to the fact that, although the gastropods lack effective dispersal mechanisms, some species can complete the colonization process by direct attachment to birds through bird activity (*Coccia et al., 2016*). The lowest richness was in LS with extensive littoral vegetation, which consisted of homogenous emergent plant communities (i.e., *T. angustifolia* community and *Phragmites australis* community), and constitutes a barrier to many species. This finding confirms those of *Harabiš (2016)*, who suggested that the vegetation should be kept scattered and prevented extensive overgrowth. In addition, fish may have an impact on community differences, and fish are often more abundant in more mature wetlands. The abundance and quantity of macrozoobenthos can be regulated because of the existence of fish, for example, Corixidae are particulary easy to subject to fish predation because they swim to the water surface for fresh air (*Schilling, Loftin & Huryn, 2009*). Furthermore, the high relative abundance of GC in the LS might have been a consequence of the surface sediments of LS being richer in organic matter.

## Seasonal dynamics

In our study, the macrozoobenthos communities of the IS and MS were significantly affected by seasonal variation, while the community of the LS was not. This result suggested that the newer subsidence wetlands showed significantly seasonal dynamics, possibly because the ecosystem was less stable. For the LS, which had broad coastal vegetation, the area and depth of water was great enough to enhance strong adaptability to seasonal environmental variations. A change in water quantity is one major factor that affects the seasonal variation of the macrozoobenthos community (*Chen et al., 2018*). Because of the low rainfall in spring, the MSWs had the lowest area and shallowest water depth in spring. Pond deoxygenation and high temperature conditions, accompanied by a drastic reduction in the pond surface area and depth and high concentrations of solutes (*Waterkeyn et al., 2008*; *Pérez-Bilbao, Benetti & Garrido, 2015*), are indicative of the general impoverishment of pond environmental conditions. Rainfall generally increased during summer. This increase may have contributed to the gradual recovery of water quantity in the MSWs.

## Species composition and environmental variables

The high biodiversity of water bodies is because individual ponds usually support different communities (*Williams et al., 2004*), even in water bodies that are close to each other, such as the MSWs in this study. The differences in the distribution of macrozoobenthos can be explained by local conditions in the wetlands (*Kitagawa, 1978*; *Arunachalam et al., 1991*; *Nelson & Lieberman, 2002*; *Nelson, 2011*; *Van Der Linden et al., 2012*). The time of subsidence has led not only to changes in the macrozoobenthos community of the newly formed wetlands but also to the internal and external characteristics of the newly formed wetlands. The water quality of the wetlands has been shown to affect the diversity of macrozoobenthos (*Lupi, Rocco & Rossaro, 2013*; *Perissinotto, Bird & Bilton, 2016*). This is especially true for macrozoobenthos in wetlands in agricultural landscapes, where extreme changes in the parameters of the aquatic environment often occur (*Meyer, Davis & Dvorett, 2015*), which in turn, affect the macrozoobenthos communities.

   In our study, the ORP, area, DO, and TDS of water had the greatest impact on the macrozoobenthos community (Fig. 6). ORP can reflect the DO status of the water, and the macrozoobenthos community varied significantly with the DO of the water (*Connolly, Crossland & Pearson, 2004*). If DO of the water is at a low level, DO can become a limiting factor for macrozoobenthos in the habitat (*Martien & Benke, 1977*; *Beisel, Usseglio-Polatera & Moreteau, 2000*; *Allan & Castillo, 2007*). For instance, in our study, *Polypedilum* sp. increased with increasing DO of the water, while *Bellamya purificata* was negatively related to the DO of the water (Fig. 6), showing their ability to tolerate and thrive in this type of water environment (*Ocon, Rodrigues Capitulo & Paggi, 2008*). The water area of the same coal mining subsidence area will gradually expand with the subsidence time. The area of the wetland has been shown to have a significant impact on the diversity and richness of the taxa (*Studinski & Grubbs, 2007*).
For example, in our study, *L. hoffmeisteri* increased with increasing water area, while *Gyraulus convexiusculus* and *Tanytarsus* sp. were negatively related to the water area. This relationship for *G. convexiusculus* and *Tanytarsus* sp. might help explain their decline during LS, as the water area increased over this period (Fig. 6). Moreover, the area of the pond can affect the temperature of the water as well as the interaction between competition and predation (*Pearman, 1994*). TDS, which is a mixture of salts (e.g., magnesium, sodium, calcium, potassium, chlorides, bicarbonates, and sulphates), organic matter and other dissolved materials in water (*Weber-Scannell & Duffy, 2007*). In our study, an increase of TDS affected most species negatively (Fig. 6), confirming the findings of many studies (*Ndaruga et al., 2004*; *Timpano et al., 2015*; *Mwedzi, Bere & Mangadze, 2016*). TDS also causes toxicity through increases in Sal, changes in the ionic composition of the water and toxicity of individual ions (*Weber-Scannell & Duffy, 2007*).

## CONCLUSIONS

We found that the macrozoobenthos community in the newly formed wetlands showed high diversity, and the developmental trajectory of the macrozoobenthos community was significantly affected by the time-lapse of the subsidence, showing the continuous succession of the diversity of the newly formed wetland during an

intermediate time period. The stage is the highest and then gradually decreases. We believe that in the early stages of newly formed wetlands, the diversity of the macrozoobenthos community is mainly related to the life history and migration ability of animals. As time proceeds, the changes in the environmental factors gradually play a role, and these changes are mainly driven by the subsidence time.

The drastic changes in the surface structure caused by the subsidence of coal mining have placed enormous pressure on both the ecosystem and the environment. However, our findings have challenged the original understanding of coal mining subsidence. We believe that we can see not only some secondary geological disasters that may occur in coal mining subsidence but also ecological opportunities introduced by the formation of new wetlands in coal mining subsidence areas. The new habitats in post-mining areas signify the diversity of (micro-) habitats across a broad spectrum of succession stages. These special secondary habitats have been strengthened due to the decreasing natural habitats. Therefore, secondary habitat geography should be an integral part of nature conservation. We should not only protect secondary habitats but also protect their continuous succession process.

## APPENDIX

| The species list of macrozoobenthos of three stages of mining subsidence wetlands. | | | |
|---|---|---|---|
| Taxa | Initial stage | Middle stage | Late stage |
| **Nematoda** | | | |
| Nematodae sp. | 3 | | 16 |
| **Annelida** | | | |
| **Oligochaeta** | | | |
| Limnodrilus hoffmeisteri | 17 | 103 | 1,188 |
| Branchiura sowerbyi | 36 | 176 | 24 |
| **Hirudinea** | | | |
| Glossiphonia lata | | 1 | 1 |
| Whitmania pigra | 4 | 2 | 3 |
| **Mollusca** | | | |
| **Gastropoda** | | | |
| Cipangopaludina chinensis | | 4 | |
| Bellamya purificata | | 15 | 2 |
| Gyraulus convexiusculus | 8 | 131 | |
| Gyraulus compressus | 5 | 140 | |
| Parafossarulus striatulus | | 19 | 1 |
| Physa acuta | 58 | 85 | |
| Radix lagotis | 123 | 315 | 2 |
| Radix swinhoei | 4 | | 1 |
| Galba pervia | 16 | 35 | |
| Galba turncatula | | 2 | |
| Alocinma longicornis | | 1 | 1 |

(Continued)

| Taxa | Initial stage | Middle stage | Late stage |
|---|---|---|---|
| *Polypylis hemisphaerula* | | 6 | |
| *Hippeutis cantori* | 52 | 105 | |
| *Hippeutis umbilicalis* | 5 | 24 | |
| **Arthropoda** | | | |
| **Insecta** | | | |
| **Diptera** | | | |
| *Chironomus plumosus* | 2,431 | 986 | 730 |
| *Hydrobaenus* sp. | | 1 | |
| *Stictochironomus* sp. | | 1 | |
| *Einfeldia* sp. | 769 | 383 | 123 |
| *Dicrotendipes* sp. | 25 | 22 | 1 |
| *Orthocladius* sp. | 12 | 20 | |
| *Psectrocladius* sp. | 1 | 3 | |
| *Paratendipes* sp. | 1 | | |
| *Chaetocladius* sp. | 2 | | |
| *Cladopelma* sp. | 2 | 8 | |
| *Diamesa* sp. | | 1 | |
| *Tvetenia* sp. | 7 | 120 | |
| *Propsilocerusi* sp. | 67 | 37 | 426 |
| Chironmidae sp. | 12 | | |
| *Tanytarsus* sp. | 57 | 21 | 1 |
| *Glyptotendipes* sp. | 2,343 | 2,034 | 272 |
| *Cricotopus* sp. | 17 | 41 | 21 |
| *Endochironomus* sp. | | 3 | |
| *Polypedilum* sp. | 420 | 148 | 46 |
| *Cryptochironomus* sp. | | | 1 |
| *Clinotanypus* sp. | 1 | | |
| *Procladius* sp. | 66 | 142 | 7 |
| *Ablabesmyia* sp. | | 2 | |
| *Tanypus* sp. | 69 | 5 | 109 |
| *Culicoides* sp. | 1 | | 12 |
| *Musca domestica* | | 1 | 13 |
| Tipulidae sp. | | | 4 |
| **Coleoptera** | | | |
| Hydrophilidae sp. | | 2 | |
| *Cybister chinensis* | 1 | 5 | |
| *Laccophilus difficilis* | 1 | 5 | 1 |
| *Heterocerus sauteri* | | 8 | |
| *Chlaenius* sp. | | 1 | |
| Carabidae sp. | | 1 | |
| **Trichoptera** | | | |
| *Phryganea* sp1. | | 1 | 6 |
| *Phryganea* sp2. | 1 | | |

| Taxa | Initial stage | Middle stage | Late stage |
|---|---|---|---|
| (continued). | | | |
| **Odonata** | | | |
| *Gomphus* sp. | 4 | 3 | 2 |
| *Anax* sp. | 3 | | |
| *Aeshna* sp. | 1 | 7 | |
| *Anisogomphus* sp. | 8 | 3 | 1 |
| *Libellula* sp. | 5 | 10 | 2 |
| *Cercion* sp. | 61 | 72 | |
| *Lestes* sp. | | 4 | |
| *Ischnura Iabata* | | 4 | |
| *Aciagrion hisopa* | 9 | 47 | |
| **Ephemeroptera** | | | |
| *Baetis* sp. | 14 | 101 | 1 |
| *Siphlonurus* sp. | | 2 | |
| **Hemiptera** | | | |
| *Diplonychus esakii* | 3 | 1 | |
| Corixidae sp. | 45 | 102 | |
| **Crustacea** | | | |
| **Decapoda** | | | |
| *Palaemonetes sinensis* | 1 | | 1 |

## ACKNOWLEDGEMENTS

The authors thank Fengyue Shu, Chenglong Ma, Yuehuan Dong, Jingtai Li, Guangshuo Han, Mengjie Zhang, Lilei Zhou, Shuaikai Wu, Kuo Sun, Yuxing Hu, and Guofeng Yang for field assistance.

## Funding

This work was supported by the Foundation of State Key Laboratory of Coal Mine Disaster Dynamics and Control, Chongqing University (2011DA105287—ZD201402). The funders had no role in study design, data collection and analysis, decision to publish, or preparation of the manuscript.

## Grant Disclosure

The following grant information was disclosed by the authors:
Foundation of State Key Laboratory of Coal Mine Disaster Dynamics and Control, Chongqing University: 2011DA105287—ZD201402.

## Competing Interests

The authors declare that they have no competing interests.

## Author Contributions

- Guanxiong Zhang performed the experiments, analyzed the data, contributed reagents/materials/analysis tools, prepared figures and/or tables, authored or reviewed drafts of the paper.
- Xingzhong Yuan conceived and designed the experiments, contributed reagents/materials/analysis tools, authored or reviewed drafts of the paper, approved the final draft.
- Kehong Wang analyzed the data, contributed reagents/materials/analysis tools, authored or reviewed drafts of the paper.

## Field Study Permissions

The following information was supplied relating to field study approvals (i.e., approving body and any reference numbers):

Field experiments were approved by the Foundation of State Key Laboratory of Coal Mine Disaster Dynamics and Control, Chongqing University (project number: 2011DA105287—ZD201402).

## Data Availability

The raw measurements are available in the Supplementary Files, including species data and environmental data. The species data shows the abundance of each species on four sampling times and three subsidence histories.

## Supplemental Information

Supplemental information for this article can be found online at http://dx.doi.org/10.7717/peerj.6456#supplemental-information.

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
