# Peer review of "Biodiversity and temporal patterns of macrozoobenthos in a coal mining subsidence area in North China"

_PeerJ, doi:10.7717/peerj.6456_

## Round 0.1 · original submission · Major Revisions

Both reviewers found this manuscript to be well written and of interest to the field. The reviewer comments mostly concern areas that the authors should further clarify in their revision. Also, please ensure that all abbreviations used in figures 2-4 are defined in the figure legends so that readers don't need to search for the meaning of the abbreviations in the text.

·

Basic reporting

Overall this is a well-written ms testing clear hypotheses with well-organized structure. The introduction and background clearly set up and justify the study. However, the results section needs significant clarification, and the methods also need some minor clarifications. The raw data only for the the species abundances are included, however, the environmental data has not been made available. Additionally there are a number of minor comments which need addressing, listed in the general comments section.

Experimental design

This ms fits the aims and scope of the journal, and the research question is clearly defined. The analyses seem appropriate, however, I have highlighted some points in the general comments to the author that need to be addressed.

Validity of the findings

The validity of the findings appears to be sound, but clarification in the methods, and a re-written results section is needed to fully evaluate the discussion of the findings.
Data appear to be robust, however, as mentioned in the basic reporting, the environmental data are not included, and there is some confusion about the number of samples and aggregation of environmental data.

Additional comments

1. Your most important issue is in the results section. There are a number of places where percentages are used, but it is unclear what they are referring to. e.g. L199-200: "...molluscs accounted for 20.59% of the community..." Is this percent referring to numerical density, species richness, biomass? The statistical test also needs to be clarified in the text. e.g. L 206 - I assume this is this Tukey HSD test? please clarify. There are also some typos and misstated results. e.g. L 225, I believe that "abundance" should actually be "richness".
I recommend including a table of results to help clarify this.

2. Your second most important item is some clarification needed in the methods. Particularly the index of relative importance subsection, and a clarification of the environmental variables used in the CCA.

Specific comments:
L55-56. I disagree with these characteristics applying to all macrozoobenthos. Particularly, the "long life" and "slow movement". Many insect taxa have short generations and have active adult dispersal phases, making them able to rapidly colonize habitats, as you point out in the discussion (L295)
L 107. Species names should be italicized.
L 130-132. Please clarify "broad coastal vegetation". Does this include the 2 species you list, or does it mean something different?
L 133. What is meant by "homogeneous monogenic plant community"? Is this referring to macrophytes, riparian or emergent vegetation?
L 138-139. Please clarify your sampling of environmental variables. If they were only collected in 2 seasons, are the results in table 1 an average between the 2 sampling times? Was there any significant seasonal variation in environmental variables?
L 147. Clarify lowest possible classification unit. e.g. Species or Genus for everything? Species for some groups, other taxonomic level (e.g. Family) for other groups?
L 163-169. Here you describe the equation for the index of relative importance, but I do not see any results of these analyses, or discussion of them. Was this analysis used? If it was, it needs to be clearly articulated in the results section. If not, this subsection needs to be removed.
L179. Here it states that the relationship between the community and environmental factors were assessed using CCA. Which environmental factors were assessed? Table 1 contains 11 variables, but figure 5 only has 4 variables on the plot. Were these 4 variables determined via some model selection? If so, that process needs to be explained. Or if they were the only variables used, this needs to be clearly stated and justified in the methods.
L 192-195. Were only the abundance data averaged? What about biomass and richness data? It's unclear exactly what you're testing here. e.g. "... abundance data were averaged from three ponds and then averaged from four sampling seasons"

Results: as mentioned above, this section needs significant clarification. Please clearly indicate exactly what the percentages are referring to, and which statistical tests are being conducted. I would also recommend clearly defining terms and making them consistent throughout results. For example you mention abundance and density. Are these referring to the same things? Or absolute abundance and relative density?

L 199-202. You state that the main group is molluscs with 14 species and 20.59% of the community, and then say that insects had 48 species and 70.59%. Shouldn't insects be the "main" group then? Or are you defining molluscs as the main group for some other reason?
L 219. I recommend replacing "P=0.000" with "P < 0.001".
L 220. What does "the highest record" refer to?
L 225. You state that the abundances significantly vary, but the P-value for GC is 0.444.
L 227. What does the "highest values" refer to? A specific FFG, or total abundance or richness?
L 233. a P-value of 0.995 for the PR FFG indicates they are not affected.
L 236. Recommend changing sentence to" "Significant changes in SC occured only between MS and LS, where relative densities in MS were significantly higher that those in LS.
L 246. Please clarify which environmental factors were used, or if there was a model selection process to select only the 4 listed in Figure 5.
L 248-249. Is "environmental factor" supposed to be plural? e.g. all factors?
L 246-253. I have a hard time following the results here. Granted, I have little experience with CCA, but a brief explanation of what the eigenvalues, and correlation coefficients represent, and how the percent explanation of variance for the 4 factors were calculated would be helpful.
L 273. Clarify what is meant by "irreversible degraded state".
L 275. What other secondary habitats are you referring to?
L 277. Are there few natural habitats because they have been removed from land use changes, or are there few in general because of homogeneous habitat?
L 277. How are "low-quality ponds" defined
L 298. Clarify what higher density refers to. E.g. higher density than what, LS or IS MSWs?
L 315. there is an extra period after the Cristina et al. 2016 reference.
L 315-317. I don't understand what this sentence means.
L 327-328. This sentence is circular and doesn't make sense. E.g. what is the effect of seasonal changes, and what is it exacerbated by?
L 334. WATERKEYN reference appears to be wrong (authors are all capitalized)

Reviewer 2 ·

Basic reporting

No comment.

Experimental design

No comment.

Validity of the findings

No comment.

Additional comments

The paper presents the diversity and temporal patterns of macrozoobenthos in coal mining subsidence wetlands. It is a topic of interest to the researchers in the related areas. The findings are interesting to a wide audience and of importance for future development of biodiversity conversation in post-mining landscape. The paper provided sufficient introduction to concepts and citation of relevant literature, and was written in a clear and concise language.

Line 28, 29, 105- change ‘hectares’ to ‘ha’.
Line 65- references should be added in this line.
Line 92- the paper misses a citation for Figure 1. Please add a citation for Figure 1 in this paragraph.
Line 107- two plant species in this line should be italicized.
Line 188- ‘H'’ should be italicized.
Line 219- change ‘subsidence’ to ‘the time of subsidence’.
Line 284, 288, 299- since the paper mainly focused on the macrozoobenthos, the macrozoobenthos list table(at least to famlily level) is very necessary as online APPendix. Please add a citation for the macrozoobenthos list table in this line.
Line 302- change ‘farmland’ to ‘i.e. farmland’.
Line 315- ‘. .’, remove first ‘.’.
Line 317- references should be added in this line.
Line 336- the author described the effect of seasonal precipitation on the water quantity of MSWs. I think the author should add the introduction of precipitation season in the 90 Materials and methods (i.e. Study area).
Some of the figures should be beautified.

---

## Round 0.2 · Minor Revisions

Reviewer 1 has just a few additional minor comments for the authors to address.

·

Basic reporting

The authors have clearly addressed all of my comments and concerns, and I believe that the manuscript is now suitable for publication in PeerJ. However, I have noted a few minor comments below which should be addressed.

Experimental design

No comments

Validity of the findings

No comments.

Additional comments

I recommend adding some description to the figures. Namely clearly indicating what the letters in Figs. 3-5 represent (e.g. Tukey HSD, PERMANOVA, etc).

Please double check that all of the references are correct and consistently reported. e.g. Ref in L 571-573 has title in all caps.

I think there is a small typo in L 405. e.g. "We believe that can we see..."

---

## Round 0.3 · accepted · Accept

Thank you for your careful attention to the reviewer comments.

#